Inter-domain microbial diversity within the coral holobiont Siderastrea siderea from two depth habitats

http://orcid.org/0000-0002-9823-6761 Bonthond Guido 1 2
http://orcid.org/0000-0002-2811-3002 Merselis Daniel G. 1
http://orcid.org/0000-0002-6951-5390 Dougan Katherine E. 1
Graff Trevor 3
Todd William 3
http://orcid.org/0000-0002-0811-8500 Fourqurean James W. 1
Rodriguez-Lanetty Mauricio 1 rodmauri@fiu.edu
1 Department of Biological Sciences, Florida International University , Miami, FL , USA
2 Aquatic Microbiology, Institute for Biodiversity and Ecosystem Dynamics, University of Amsterdam , Amsterdam , The Netherlands
3 NASA Johnson Space Center , Houston, TX , USA
Breitbart Mya
Electronic publication date: 2018 Feb 9
Publication date: 2018
Volume: 6
Electronic Location ID: e4323
Received 2017 Aug 16; Accepted 2018 Jan 13
Copyright: © 2018 Bonthond et al.
Copyright year: 2018
Copyright holder: Bonthond et al.
License: This is an open access article distributed under the terms of the Creative Commons Attribution License, which permits unrestricted use, distribution, reproduction and adaptation in any medium and for any purpose provided that it is properly attributed. For attribution, the original author(s), title, publication source (PeerJ) and either DOI or URL of the article must be cited.
License URL: https://creativecommons.org/licenses/by/4.0/

Keywords: Symbiosis, Coral, Fungi, Microbial community, Coral-associated microbiome, Symbiodinium

Funding: NEEMO mission Florida International University This work was funded by the NEEMO mission and by a Florida International University start-up grant awarded to Mauricio Rodriguez-Lanetty. There was no additional external funding received for this study. The funders had no role in study design, data collection and analysis, decision to publish, or preparation of the manuscript.

==============================
Corals host diverse microbial communities that are involved in acclimatization, pathogen defense, and nutrient cycling. Surveys of coral-associated microbes have been particularly directed toward Symbiodinium and bacteria. However, a holistic understanding of the total microbiome has been hindered by a lack of analyses bridging taxonomically disparate groups. Using high-throughput amplicon sequencing, we simultaneously characterized the Symbiodinium, bacterial, and fungal communities associated with the Caribbean coral Siderastrea siderea collected from two depths (17 and 27 m) on Conch reef in the Florida Keys. S. siderea hosted an exceptionally diverse Symbiodinium community, structured differently between sampled depth habitats. While dominated at 27 m by a Symbiodinium belonging to clade C, at 17 m S. siderea primarily hosted a mixture of clade B types. Most fungal operational taxonomic units were distantly related to available reference sequences, indicating the presence of a high degree of fungal novelty within the S. siderea holobiont and a lack of knowledge on the diversity of fungi on coral reefs. Network analysis showed that co-occurrence patterns in the S. siderea holobiont were prevalent among bacteria, however, also detected between fungi and bacteria. Overall, our data show a drastic shift in the associated Symbiodinium community between depths on Conch Reef, which might indicate that alteration in this community is an important mechanism facilitating local physiological adaptation of the S. siderea holobiont. In contrast, bacterial and fungal communities were not structured differently between depth habitats.

Introduction

Reef building corals harbor diverse microbial communities integral to their metabolism and physiology that are important in nutrient cycling, pathogenicity, pathogen defense, and environmental resilience (Muscatine & Porter, 1977; Rohwer et al., 2001; Lesser et al., 2004, 2007; Ritchie, 2006; Reshef et al., 2006; Wegley et al., 2007; Olson et al., 2009; Raina et al., 2009; Davy, Allemand & Weis, 2012; Ziegler et al., 2017). The interdependence of the coral host and its diverse microbiome has manifested the recognition of the coral holobiont, or the sum of all microbes and the coral host (Rohwer et al., 2001; Knowlton & Rohwer, 2003). The most well-studied symbionts, among the diverse bacteria, archaea, fungi, protista, and viruses that comprise the holobiont, are Symbiodinium, which reside intracellularly within the gastroderm. These dinoflagellates conduct photosynthesis and provide the host with fixed carbon (Muscatine & Porter, 1977). The Symbiodinium genus comprises many genetically diverse “types” or species with different phenologies that are associated with a multitude of hosts, including Foraminifera, Mollusca, Porifera, and Cnidaria (LaJeunesse, 2001; Rodriguez-Lanetty, 2003; Coffroth & Santos, 2005; Stat, Carter & Hoegh-Guldberg, 2006). The degree to which different coral species are able to associate with distinct Symbiodinium types has been termed “host flexibility” (Baker, 2003). It has been hypothesized that flexible corals are better able to adapt to changing environmental conditions (Buddenmeier & Fautin, 1993; Baker, 2003; Baird et al., 2007) and to different environmental settings along latitudinal gradients (Rodriguez-Lanetty et al., 2001; Rodriguez-Lanetty & Hoegh-Guldberg, 2003).

Recent efforts, using highly sensitive next-generation sequencing, revealed that the coral microbiome is far more diverse than previously thought (Amend, Barshis & Oliver, 2012; Lema, Willis & Bourne, 2012; Rodriguez-Lanetty et al., 2013; Bayer et al., 2013a, 2013b; Soffer, Zaneveld & Vega-Thurber, 2015; Ainsworth et al., 2015; Hernandez-Agreda et al., 2016; Brown et al., 2017). Even though understanding of the coral microbiome has developed considerably (reviewed in Rosenberg et al., 2007; Ainsworth, Vega-Thurber & Gates, 2010; Bourne, Morrow & Webster, 2016), the functional contributions of microbes associated with the host are still poorly understood. Apart from Symbiodinium and bacteria, other microbes, such as fungi, have only received occasional attention. It is, however, fairly well established that a diverse fungal community is naturally associated with the healthy coral, particularly within the skeleton (reviewed in Raghukumar, 2012; Yarden, 2014; Kendrick et al., 1982; Le Campion-Alsumard, Golubic & Priess, 1995; Domart-Coulon et al., 2004; Wegley et al., 2007; Vega-Thurber et al., 2009; Littman, Willis & Bourne, 2011; Amend, Barshis & Oliver, 2012). Symbiodinium and bacterial communities have been characterized and studied with next-generation sequencing approaches, however, this has only been applied on one occasion for fungi (Amend, Barshis & Oliver, 2012). While ecological understanding of the bacterial component of the coral holobiont is growing, the role of fungi remains understudied and enigmatic. To develop a baseline understanding of the coral microbiome ecology, it is imperative to study all microbial taxa, including fungi.

Reef building corals inhabit the upper layer of the ocean and are typically exposed to strong fluctuating environmental conditions, such as light intensity, temperature, seawater chemistry, and microbial fluxes which structure the composition of coral-associated microbial communities (Vega-Thurber et al., 2009). Corals survive over a considerable depth distribution, ranging from directly beneath the surface, to deeper and more stable mesophotic regions (Lesser, Slattery & Leichter, 2009). Microbial communities are likely key contributors to their capacity to persist in these disparate and fluctuating environments. It has been postulated that microbial succession within the microbiome could mitigate stress associated with changing environmental conditions (Reshef et al., 2006). Therefore, shifts between different microbial groups along the depth cline might support local ecological success of corals.

Given the steep decline of coral reefs globally (Gardner et al., 2003; van Hooidonk et al., 2014), an urgent need exists to develop a better understanding of the contribution of the microbiome (viruses, prokaryotes, and microeukaryotes) to holobiont resilience and adaptation. The coral selected for this study, Siderastrea siderea, is a gonochoric coral and an abundant massive coral in the Florida reef tract (Lirman & Fong, 2007). This species is considered to be among the most adaptive corals on the Caribbean reefs as it is relatively resilient to several environmental stressors (Colella et al., 2012; Castillo et al., 2014) and typified by high recovery rates from major bleaching events (Banks & Foster, 2017). Therefore, S. siderea represents a suitable species to investigate microbiome patterns between different localities on the same reef. The aim of this study was to reconstruct a detailed image of the Symbiodinium, bacterial, and fungal communities associated with S. siderea and investigate differences in microbiome diversity at the inter-domain level between two different depth habitats on the same reef. Previous studies have shown that while alpha diversity generally remains stable, coral-associated Symbiodinium and bacterial community structures (beta diversity) are variable with depth, particularly for corals occurring over a wide depth range (Klaus et al., 2007; Lee et al., 2012; Bongaerts et al., 2015; Glasl et al., 2017). For coral-associated fungal communities, the influence of depth on diversity has not yet been investigated.

We collected samples from multiple S. siderea colonies at Conch reef in the Florida Keys from two depth intervals (17 and 27 ± 1 m) and conducted high-throughput sequencing of the Symbiodinium chloroplast 23S (cp23S) hyper variable region, the V123 region of the bacterial 16S ribosomal small subunit (SSU) and the fungal internal transcribed spacer-2 (ITS2). We hypothesized that between the two depth habitats sampled in this study, S. siderea would harbor distinctive Symbiodinium and bacterial communities (i.e., beta diversity), as predicted by previous work (Klaus et al., 2007; Lee et al., 2012; Bongaerts et al., 2015; Glasl et al., 2017). We extended this hypothesis to the fungal community, reasoning that shifts in the microbiome between depths with different local environments are most likely not restricted to the Symbiodinium genus and bacterial domain alone, but act at the inter-domain level, thus including the associated fungal community. In addition, we aimed to explore shifts within the microbiome and identify depth habitat-specific microbes by performing a species indicator analysis (Caceres & Legendre, 2009) and investigate co-occurrence and mutual exclusion patterns by constructing a microbial network.

Methods

Sample collection

Siderastrea siderea colonies were sampled from 27 ± 1 m (n = 8) and 17 ± 1 m (n = 6) from Conch reef surrounding the Aquarius Reef Base (Florida International University) in the Florida reef tract, approximately 8 km offshore from Key Largo. These samples were collected by aquanauts performing a saturation mission in the Aquarius Reef Base habitat during the 20th NASA Extreme Environment Mission Operations (NEEMO) mission in July 2015. Preceding the sampling, the dive team was properly trained and familiarized with the method of collection and the operation of the sampling equipment. Diving from the saturation habitat facilitated greatly extended dives to the deeper collection areas in particular. Fragments of approximately 2 cm2 surface area were sampled from healthy colonies from 27 and 17 m using chisel and hammer. Samples were brought to the surface in seawater containing 50 mL tubes within 4 h after collection. Seawater around the sampled corals was immediately removed before fixing the samples in a sterile 20% DMSO, 0.5M EDTA NaCl-saturated solution. It should be noted that collected samples were not actively washed with sterile seawater to remove loosely attached microbes and these microbes may be likely present at background levels in our data set and contribute to the detected patterns or absence of them. Coral tissues were homogenized with high-pressure airbrushing (Rodriguez-Lanetty et al., 2013) and the resulting material was preserved in new 20% DMSO solution at −20 °C until DNA extraction.

DNA extraction, amplification, and sequencing

From each of the eight (27 m) and six (17 m) samples, DNA was isolated using the MP FastDNA spin kit for soil (MP Biomedicals, Santa Ana, CA, USA) with an additional ethanol precipitation step. All DNA samples were divided into three aliquots to allow sequencing of the bacterial 16S rRNA gene, Symbiodinium cp23S rRNA gene, and fungal ITS2 rRNA gene from the same sample.

The cp23S fragment was amplified with primers from Santos, Gutierrez-Rodriguez & Coffroth (2003; 5′-TCAGTACAAATAATATGCTG-3′ and 5′-TTATCGCCCCAATTAAACAGT-3′) and the 16S rDNA V123 region with 27F and 519R primers (5′-AGRGTTTGATCMTGGCTCAG-3′, 5′-GWATTACCGCGGCKGCTG-3′; Turner et al., 1999) using the HotStarTaq Plus Master Mix Kit (Qiagen, Hilden, Germany). The PCR programs for 16S and cp23S consisted of 3 min initial denaturation at 94 °C, 28 cycles of 30 s at 94 °C, 40 s at 53 °C and 1 min at 72 °C, and a final elongation step of 5 min at 72 °C.

To circumvent amplification of DNA from the coral host, the ITS2 region was amplified in a nested PCR using phylum-specific primers (Nikolcheva & Bärlocher, 2004). The first PCR program consisted of 2 min initial denaturation at 95 °C, 35 cycles of 30 s at 95 °C, 30 s at 57 °C and 0:45 min at 72 °C, followed by a final elongation step of 2 min at 72 °C. PCR reactions were conducted in volumes of 20 μL, containing 0.5 μM of each primer, 10 μL 2× PCR Master Mix (Promega, Madison, WI, USA) and 10 ng DNA template. In this first PCR, the universal forward primer ITS3 (5′-GCATCGATGAAGAACGCAG-3′; White et al., 1990) was combined with reverse primers specific for Ascomycota, Basidiomycota, Chitridiomycota, Oomycota, and Zygomycota (Nikolcheva & Bärlocher, 2004). The second PCR program comprised 2 min initial denaturation at 95 °C, 10 cycles of 30 s at 95 °C, 30 s at 60 °C and 30 s at 72 °C, and a final elongation step of 2 min at 72 °C. PCR reactions were identical to the first PCR program with 1 μL of 10−1 diluted amplicon as template. The final amplicons were generated by using the universal ITS3 forward primer and the universal ITS4 reverse primer (5′-TCCTCCGCTTATTGATATGC-3′; White et al., 1990). For all sequencing assays except the first round of the nested ITS2 PCR, primers contained Illumina adapters. Barcodes were located between adapter and forward primer. After amplification, samples were pooled per marker in equal proportions, purified using calibrated Ampure XP beads and DNA libraries were prepared following the Illumina TruSeq DNA library preparation protocol for 300 bp paired-end MiSeq sequencing at MR DNA (Shallowater, TX, USA). De-multiplexed reads were deposited in the SRA database under the accession number PRJNA356144.

Raw sequence reads were joined, de-multiplexed, quality filtered, denoised, purged of chimeras, clustered into operational taxonomic units (OTUs) using the nearest neighbor algorithm and classified with Mothur 1.36.1 software according to the MiSeq SOP (Schloss et al., 2009; Kozich et al., 2013; see Supplemental Information for the executed command steps in Mothur).

To classify the Symbiodinium cp23S sequences we used the sequences from Pochon, Putnam & Gates (2014), who conducted a multigene clade level analysis of the Symbiodinium genus and included the cp23S region using 33 unique sequences representing different types from currently all known clades (clades A–I) with the dinoflagellates Gymnodinium simplex and Polarella glacialis as outgroups. We aligned all sequences with MAFFT (Katoh & Standley, 2013) and formatted the dataset to be compatible with Mothur 1.36.1 (see Supplemental Information for the aligned, unaligned, and taxonomy files). All cp23S contigs were aligned against this dataset for denoising and OTU clustering. The same set of sequences was used to classify the contigs and remove sequences not classified as Symbiodinium. For the bacterial sequences, denoising and classification was conducted using the SILVA reference alignment (release 123; Quast et al., 2013). To effectively eliminate chloroplast, mitochondrial 16S rDNA, and unclassified sequences from the dataset, both the SILVA and GreenGenes (DeSantis et al., 2006) alignments were used. We selected the SILVA reference dataset to align and classify the 16S reads since it is both more recently updated (2015 compared to 2013 for the GreenGenes dataset) and of superior alignment quality (Schloss, 2010). Due to the high variance on the ITS2 region, a high quality reference alignment covering the full fungal kingdom does not exist. Instead, similarity distances were calculated from pairwise alignments. Sequences were initially classified with the User-friendly Nordic ITS Ectomycorrhiza (UNITE) database (Abarenkov et al., 2010), however, due to overall low similarity scores all OTU sequences were additionally compared against the full GenBank record with the basic local alignment search tool (BLAST; Altschul et al., 1990).

For each of the datasets, extremely rare OTUs (represented by 10 or less reads in the total dataset) were excluded from further analysis to prevent diversity inflation with false OTUs resulting from erroneous reads (Bokulich et al., 2013). Classification was conducted with Mothur’s implemented Wang-method and 100 bootstraps per sequence and using a criterion of a minimum of 60 bootstraps per sequence (Wang et al., 2007). 16S and cp23S reads were clustered into OTUs based on the traditional 3% difference criterion. Due to the relative high intragenomic variance on the fungal ITS2 region (Schoch et al., 2012), ITS2 reads were clustered into OTUs using a more conservative criterion of 5% difference.

Statistical analyses

Operational taxonomic unit tables were normalized with cumulative sum scaling and log transformed using the R package “metagenomeSeq” (Paulson et al., 2013). Shannon entropy indices were compared with one-way type-III ANOVAs. To test the hypothesis that microbial communities within the same depths are more similar to each other than between depths, non-parametric analysis of similarities (ANOSIMs) and permutational analysis of variances (PERMANOVAs) were calculated, using Bray–Curtis distances and 9,999 permutations. Non-metric multidimensional scaling (nMDS) plots were produced to visualize similarities in community structures.

To identify significant correlations between distributional patterns of microbes we constructed a co-occurrence network (Newman, 2006; Rodriguez-Lanetty et al., 2013). To reduce the complexity of the combined datasets, bacterial OTUs (bOTUs) were merged at the family level and Symbiodinium OTUs (sOTUs) at the type level. OTUs lacking family level classification were grouped by order or higher if all OTUs in that taxonomic level lacked a family classification. From the resulting table, we computed all possible pairwise spearman rank correlation values (ρ) and corresponding p values, corrected for a false discovery rate (Benjamini & Hochberg, 1995). Correlations were considered valid when p < 0.05 and ρ > |0.6|. The network was visualized with the software Gephi (Bastian, Heymann & Jacomy, 2009). The species indicator analysis (Caceres & Legendre, 2009) was computed from the same table as the network (i.e., containing bacteria merged at the family level and Symbiodinium at the type level) with the R package “indicspecies,” using the “multipatt” command, to test for significant associations to one of the depths and assigning bacterial families, Symbiodinium types and fungal OTUs (fOTUs) with p < 0.05 to be depth indicators.

Alignments for additionally investigated OTUs were constructed with MAFFT v. 7. (Katoh & Standley, 2013). Optimal evolutionary models were calculated with MrModelTest v. 2.3 and selected based on the AIC (Nylander, 2004). Maximum-likelihood analyses were conducted on the CIPRES Science Gateway portal (Miller, Pfeiffer & Schwartz, 2012), using RAxML v.8. (Stamatakis, 2014) with a 1,000 bootstrap iterations.

Results

Symbiodinium, bacterial, and fungal community composition

In total, 54 sOTUs were resolved from the 1,455,725 sequences remaining after quality filtering treatment with an average read count of 103,980 ± SD 90,086 per sample. OTUs were classified to types from clades A (7), B (26), C (6), D (1), F (2), G (2), and H (1). The remaining nine sOTUs lacked clade level classification by 60 bootstrap values or more. The overall most abundant, sOTU1 (39.5% average relative abundance) was classified to clade C, sOTU2 (21.7%) to clade A (A3) and sOTU3 (16.4%) and sOTU4 (10.2%) to clade B and type B1, respectively. sOTU5, classified as an H1 type, was the fifth most abundant sOTU (4.8%), and dominated in one of the 14 samples (Fig. 1A). sOTU6 was classified as a G2-1 type and was the sixth most abundant sOTU (3.3%). The one sOTU classified to clade D, type D1 (Symbiodinium trenchii), was rare and occupied the 24th position.

Figure 1 Relative distribution per sample of (A) the eight most abundant Symbiodinium OTUs, ordered by clade level classification, (B) the 10 most abundant bacterial phyla, and (C) the 10 most abundant fungal OTUs per sample.

The total amount of bOTUs after quality filtering was 714 from a total of 115,095 sequences with an average of 8,221 ± SD 4,926 reads per sample. The phylum Proteobacteria was the most abundant (49.2%), followed by Chloroflexi (14.7%), Cyanobacteria (7.4%), and Bacteriodetes (6.1%; Fig. 1B). The most abundant bOTU, bOTU1 (10.2%), was classified to the SAR116 clade (Alphaproteobacteria), followed by bOTU2 (5.7%) classified as Synechococcus (Cyanobacteria) and then bOTU3 (4.4%), classified to the clade SAR202 (Chloroflexi). bOTU4 (3.5%) and bOTU5 (3.5%) were classified to Candidatus Branchiomonas (Nitrosomonadaceae) and Candidatus Lariskella (Rickettsiales), respectively.

Despite using fungal-specific primers (Nikolcheva & Bärlocher, 2004), a relative high amount of ITS2 reads from other eukaryotes was retrieved. Only 22.2% of the 790,398 quality filtered ITS2 reads were fungal. Seventeen percent of the reads were from Cnidaria and the remaining reads from other eukaryotes. A group of fungi-like Opisthokonta (Mesomycetozoa) was also represented with 0.15% of the reads. Using the UNITE reference database in the Mothur pipeline, 184 fOTUs were classified to at least the fungal kingdom. However, additional comparison of the complete fOTU sequence set along the GenBank database showed 145 OTUs to yield high similarity scores with non-fungal sequences. These fOTUs were removed from the OTU table, resulting in 39 fOTUs with only similarity hits to fungi or Mesomycetozoa, constituting 141,706 reads in total and by average 10,122 ± SD 9,042 reads per sample. Most fOTUs belonged to Ascomycota (34). Other fOTUs were most similar to Basidiomycota (2), Entomorphthoromycota (1), and Mesomycetozoa (2). ITS2 similarity with sequences from GenBank was notably low (91.0 ± SD 8.2%). The overall most abundant fungus (fOTU1; 48.7% average relative abundance; Fig. 1C) was most similar (76%) to sequences belonging to Lulworthiales (Sordariomycetes). fOTUs 2 (7.2% average relative abundance) and 3 (6.9%) were most similar to sequences from Hyalorhinocladiella (73%; Ophiostomatales) and Physalospora (86%; Xylariales), respectively. A core set of 21 fOTUs, out of 39, was detected in >90% colonies, 15 of which were present in all samples.

Assemblage patterns in Symbiodinium, bacterial, and fungal communities

Differences in Symbiodinium community structure were detected between the two depths as visual in the nMDS plot (Fig. 2A) and statistically supported by ANOSIM and PERMANOVA analyses (p = 0.001 and 0.002, respectively). This was opposite for bacteria (pANOSIM = 0.450, pPERMANOVA = 0.232) and fungi (pANOSIM = 0.261, pPERMANOVA = 0.118) where differences between depths were not significant. While based on Shannon entropy indices no significant differences were detected in α-diversity between depths in any of the three microbial domains, the mean diversity tended to be higher in the deep habitat (27 m) across the communities (Fig. 3).

Figure 2 Non-metric multidimensional scaling plots with corresponding stress values of the distinct microbial groups associated with S. siderea: (A) Symbiodinium, (B) bacteria, (C) fungi.

Polygons are drawn around samples to visualize from which depth habitat they were collected, with 27 m (1–8) transparent and 17 m (9–15) opaque.

Figure 3 Boxplots of Shannon entropy indices for diversity in microbial groups associated with S. siderea: (A) Symbiodinium, (B) bacteria, (C) fungi.

The thick line represents the median value and boxes show the first and third quartiles. Lower and upper whiskers extend from the boxes to the extreme values or 1.5 times the inter quartile range when the extreme values are outside the range.

Co-occurrence patterns and depth indicators

The resolved network, conducted to explore co-occurrence patterns, contained 30 nodes (fOTUs, Symbiodinium types or bacterial families) and 21 edges (significant spearman correlation values above 0.6 between nodes), three of which were negative (Fig. 4). The average degree (the average number of edges per node) was 1.4 and the average weighted degree, weighted based on spearman correlation values, was 0.94. The modularity of the network or degree to which a network tends to unfold into modules or sub-communities (Blondel et al., 2008) was 1.04.

Figure 4 Co-occurrence network of S. siderea associated microbes.

Nodes are colored based on taxonomy and green edges represent positive correlation values whereas red edges indicate negative correlations.

Within the network, one large module of co-occurring nodes was identified, containing six nodes. The module was connected with three negative edges to a node classified as S-70 (Planctomycetes). The node in the module with the most edges was an uncultured bacterial group that is designated in the SILVA database as “PAUC34f” and had three positive correlations and one negative correlation with the Planctomycete group. Other nodes were classified to Proteobacteria (Desulfurellaceae, Nitrospinaceae, and Nitrosococcus), Chloroflexi (SAR202), and Acidobacteria (Subgroup 11). In the overall network, the majority of nodes were bacterial. Symbiodinium type A13 and an unclassified clade A type co-occurred in a pair. Fungal nodes in the network were fOTUs 23 and 33, most similar to Infundibulomyces (Ascomycota; incertae sedis) and Conicomyces (Chaetosphaeriales), respectively. fOTU 23 co-occurred with the Bacteriodetes family Flammeovirgaceae and fOTU33 with Rhodobiaceae (Alphaproteobacteria; Rhizobiales).

The species indicator analysis identified Symbiodinium types C1 and C90 as indicators of the deep habitat (27 m), (Table 1). Among the bacterial groups, RSB-22 (Verrucomicrobia) was indicative for 27 m and an unclassified alphaproteobacterium and Desulvovibrio (Deltaproteobacteria) were for 17 m. fOTU 12 (with 85% ITS2 similarity to a sequence from Basidiobolus, phylum: Entomorphthoromycota) was significant for 17 m. None of the depth indicators appeared in the network.

Table 1 Depth indicators identified in this study.

	Phylum/clade	Depth (m)	p Value	
Bacteria	
 RSB-22	Verrucomicrobia	27	0.031	
 Unclassified α-proteobacterium-III	Alphaproteobacteria	17	0.046	
 Desulfovibrio	Deltaproteobacteria	17	0.049	
Symbiodinium	
 C90	Clade C	27	0.001	
 C1	Clade C	27	0.037	
Fungi	
 fOTU12	Entomorphthoromycota	17	0.041	
Note:

Symbiodinium types, fungal OTUs, and bacterial families significantly associated to one of the depth habitats (17 and 27 m).

Discussion

Different coral-associated microbial communities have been characterized with the use of high-throughput sequencing (e.g., Symbiodinium (Thomas et al., 2014; Quigley et al., 2014; Arif et al., 2014; Green et al., 2014; Klepac et al., 2015; Lucas et al., 2016; Cunning, Gates & Edmunds, 2017), bacteria (reviewed in Rosenberg et al. (2007), Ainsworth, Vega-Thurber & Gates (2010), and Bourne, Morrow & Webster (2016)), and fungi (Amend, Barshis & Oliver, 2012)). This study is the first to characterize these subcomponents of the coral microbiome simultaneously. However, our sampling was limited to only a small set of 14 colonies from two depths on one reef and although it is reasonable to assume that light conditions are considerably different between 10 m depth, the data are not supplemented with measurements of environmental parameters to verify this assumption and identify potential other differences between sites.

Siderastrea siderea associated Symbiodinium

Along the depth cline, light intensity decreases exponentially and is a major variable structuring reef diversity composition (Van den Hoek et al., 1978). In addition to overall intensity, longer wavelengths attenuate faster and as a result, the light spectrum narrows with depth. It is therefore not surprising that changes in the dominantly associated Symbiodinium type have been shown to occur over relative short depth ranges (Frade et al., 2008; Bongaerts et al., 2015). Despite the narrow range covered in this study, an exceptional diversity of Symbiodinium was detected in association with S. siderea, though it is not the first work to report high diversity of Symbiodinium associated with S. siderea (Finney et al., 2010; Bongaerts et al., 2015). Contrastingly, other work has indicated limited diversity of Symbiodinium hosted by S. siderea, though the dominant symbiont may differ (Baker, 2001; Warner et al., 2006). Siderastrea congeners have been previously shown to follow a similar reef site driven pattern of low diversity within site, but with differences between sites (Thornhill, Fitt & Schmidt, 2006; Monteiro et al., 2013).

Depth zonation in S. siderea associated Symbiodinium was reported by Bongaerts et al. (2015) where, Symbiodinium D1 was dominant at 2 m depth. For the site sampled in this study, such shallow depths are unattainable and were not included. However, sOTU24, which was classified as D1, was detected at background levels (0.012% average relative abundance) and could potentially have been more abundant at shallower depths. From 2 to 10 m in depth Bongaerts et al. (2015) detected clade B types to be dominant in a minority of sampled colonies and from 5 to 50 m the type C3 was the predominantly detected primary symbiont. Likewise, in this study we identified an unclassified clade B (sOTU3) and clade C (sOTU1) type to be the most abundant Symbiodinium types at 17 and 27 m, respectively. Although the abundance of sOTU1 was by average more than two-fold higher at 27 m compared to 17 m on average, sOTU1 (being the only unclassified clade C type) was not detected as an indicator of 27 m. However, the other clade C OTUs, C90 (sOTUs 34, 45, and 52) and C1 (sOTUs 38 and 53) were indicators of the 27 m depth.

It should be noted that we did not test for a genotype effect and that the detected Symbiodinium zonation depth patterns might reflect the existence of S. siderea genotypes specific associations. Nevertheless, to our knowledge there is no published data indicating that genotypes within coral species living under the same conditions host Symbiodinium symbionts from differing clades while on the other hand depth has been shown to influence coral-associated microbes (hence Bongaerts et al. (2015) but see also Klaus et al. (2007), Frade et al. (2008), Lee et al. (2012), and Glasl et al. (2017)).

In addition to hosting a diverse background community, varying Symbiodinium types from different clades can be the dominant colonizer. Therefore, we posit that Siderastrea have the physiological capacity for broad host flexibility, but suggest that realized flexibility is winnowed by environmental conditions. Stable conditions could select for a community strongly dominated by one type. The observation of high Symbiodinium diversity may in part result from our study being the first to employ high-throughput sequencing of S. siderea associated Symbiodinium, which has the power to detect the presence of rare symbionts. However, many of the S. siderea symbionts are present in sufficient abundance for detection through the previously employed methodologies such as cloning and DGGE. This then suggests that the historically limited sampling effort on this host species and/or a novelty specific to our study site explain the surprising diversity we report.

Symbiodinium type A3, identified as the second most abundant sOTU, is in the Caribbean known for predominance in shallow specialist hosts, including Acropora and branching Porites from 1 to 10 m (LaJeunesse, Loh & Trench, 2009). Clade A types have physiological adaptations to facilitate light tolerance (Banaszak, LaJeunesse & Trench, 2000; Reynolds et al., 2008). Despite this, Finney et al. (2010) also reported an increasing abundance of A3 with increasing depth within Siderastrea radians and Stephanocoenia intersepta hosts. While Siderastrea spp. are recognized to be more efficient at scattering light than Acropora (Marcelino et al., 2013), it seems unlikely that S. siderea from the deeper habitat are able to mimic the light environment of corals more than 10 m shallower. Thus, the A3 identified in deep Siderastrea and Stephanocoenia may represent a cryptic type or phenotypically divergent genotype as elsewhere in the A3 lineage (Parkinson et al., 2015; Pinzon et al., 2011).

To our knowledge, this is only the second study to report high abundance of Symbiodinium clade H from a scleractinian (after Bongaerts et al., 2015) and one of few to report clade G (van Oppen, Mahiny & Done, 2005; Kimes et al., 2013; Thomas et al., 2014). These clades are known to associate with foraminifera and porifera, respectively (Coffroth & Santos, 2005; Stat, Carter & Hoegh-Guldberg, 2006; Pochon, Putnam & Gates, 2014). Both H1 and G2 Symbiodinium types were present in all 14 samples and the H1 OTU was the dominant sOTUs in one of the deep habitat samples. Similarly, Bongaerts et al. (2015), detected H1 as the dominant Symbiodinium type in one Porites astreoides colony. However, further experimental study is required to characterize the nature of these associations.

Siderastrea siderea associated bacteria

The composition of the bacterial community associated to S. siderea was not structured differently between the depths covered in this study, nor did we detect differences in total Shannon diversity. The stability in alpha diversity is in line with previous environmental sequencing studies that sampled coral microbiomes from different depths (Klaus et al., 2007; Lee et al., 2012; Glasl et al., 2017).

The 16S sequence of the most abundant bacterial phylotype (bOTU1) was classified to the SAR116 clade which despite its classification under the Rickettsiales in the SILVA record, is part of the Rhodospirillaceae (Alphaproteobacteria; Rhodospirillales; Grote et al., 2011). Further investigation of this bOTU showed it was 99% similar to 16S sequences from coral holobionts (Lopez et al., 2008; Sunagawa, Woodley & Medina, 2010; Yang et al., 2013), and identical to that of a bacterium which was recently shown to be a calcifying endosymbiont of a sponge, where it lacked a cell wall and was vertically transmitted (Uriz et al., 2012; Garate et al., 2017). Garate et al. (2017) opted the presence of a “calcibacterium” clade within SAR116 mainly including sequences from sponge and coral holobionts. Based on phylogenetic analysis including sequences from Oh et al. (2010) and Garate et al. (2017) as well as all SAR116 sequences in the SILVA record (Fig. S1), bOTU1 was resolved within a highly supported clade within SAR116 that was indeed dominated by sequences from sponges and corals. As discussed in Garate et al. (2017), the presence of bacterial calcite spherules has not yet been investigated in cnidarians but the high abundance of a calcibacterium related bOTU within the S. siderea microbiome and repeated recovery of related sequences from other corals (Lopez et al., 2008; Sunagawa, Woodley & Medina, 2010; Yang et al., 2013), set the stage for further research to explore whether bacterial calcification might occur in corals as well.

The second most abundant bOTU was classified to Synechococcus, a group of cyanobacteria found in other studies as an abundant coral symbiont that fixes nitrogen utilized by Symbiodinium (Lesser et al., 2004, 2007; Carlos, Torres & Ottoboni, 2013; Morrow et al., 2015). Due to the use of different 16S rDNA regions in this study and Lesser et al. (2004), sequences cannot be directly compared. Nevertheless, the high abundance of Synechococcus in S. siderea might suggest a potential contribution of nitrogen fixation within this holobiontic association, similar to that of Montastraea cavernosa and Synechococcus (Lesser et al., 2004). Another high abundant bacterium (bOTU3) was classified to Chloroflexi SAR202. Chloroflexi are not commonly reported in coral microbiomes, however have been detected on a few occasions (Kimes et al., 2013; Lee et al., 2012), and in particular from S. siderea (Cardénas et al., 2012; Kellogg et al., 2014). Finally, the abundant bOTU5 was assigned to the Rickettsiales, and further phylogenetic investigation showed this sequence to cluster with sequences acquired from other Cnidaria (Fraune & Bosch, 2007; Sunagawa, Woodley & Medina, 2010; Kimes et al., 2013) under a fully supported node (see Fig. S2), sister to the vertically transmitted intracellular Candidatus Lariskella (Matsuura et al., 2012) and Candidatus Midichloria, a candidate genus of ecologically wide spread intracellular symbionts originally isolated from Acanthamoeba (Montagna et al., 2013).

Siderastrea siderea associated fungi

Although fungi have been studied primarily in terrestrial ecosystems, it is known that their presence in the coral holobiont is ubiquitous (Kendrick et al., 1982; Le Campion-Alsumard, Golubic & Priess, 1995; reviewed in Yarden (2014) and Raghukumar (2012)). The ecology of the coral-associated fungi, however, is poorly understood (Golubic, Radtke & Le Campion-Alsumard, 2005). While fungi within the skeleton (or endolithic fungi) may invade coral tissue during times of stress (Yarden, 2014), fungi have also been found to parasitize on endolithic algae (Le Campion-Alsumard, Golubic & Priess, 1995).

The overall low ITS2 similarity of fOTU sequences to GenBank (91.0 ± SD 8.2% average similarity) corroborates the presence of a high novel diversity associated with S. siderea’s microbiome. This agrees with Amend, Barshis & Oliver (2012), where Acropora hyacinthus associated fungi had an average similarity of 97% to sequences from GenBank, using the more conserved large ribosomal subunit (LSU) rDNA marker. A notable pattern within the fungal community was the dominance of the Lulworthiaceae related fOTU1 as the most abundant in nine of the 14 samples. In each of the remaining samples, different fungi (fOTUs 2–6) occupied this position.

Amend, Barshis & Oliver (2012) did not detect differences between fungal communities of A. hyacinthus from sites on the same reef that differed by 1.5 °C. Over this range the dominant Symbiodinium type shifted. Similarly, in this study where samples were collected from the same reef but with difference in depth instead of temperature, the dominant Symbiodinium type shifted (from the unclassified C type to a combination of B types) while such a pattern was not detected for fungal communities.

In Amend, Barshis & Oliver (2012) a core set of 11 OTUs was identified, present in >90% of the samples. Notably, 21 fOTUs in our study were detected above this threshold. Due to the use of different markers, sequences of these core-members cannot be compared between studies. However, Hortaea werneckii and Lindra sp., highlighted in their study, are related to core fOTUs from S. siderea. Among the well-classified fOTUs, fOTU35 is highly similar to H. werneckii (99%), present in 13 of the 14 samples. Lindra belongs to the Lulworthiales, of which 7 fOTUs (4 in more than 90% of the samples) yielded closest similarity hits including the most abundant fOTU1 and fOTU2. The Lulworthiales is an order of marine fungi, predominantly isolated from algae or detritus (Kohlmeyer, Spatafora & Volkmann-Kohlmeyer, 2000). Given the low similarity, the sequences acquired from S. siderea likely reflect one or more novel groups within or related to this order.

The Internal transcribed spacer region has become the standard DNA marker in fungal environmental sequencing assays (Abarenkov et al., 2010). Despite this, our data emphasize that coral-associated fungal ITS sequences are underrepresented in sequence databases, resulting in poor classification of detected OTUs. This problem was also encountered by Wainwright et al. (2017) and corroborates that marine fungi are poorly studied with respect to their terrestrial relatives (Richards et al., 2012; Peay, Kennedy & Talbot, 2016). For future attempts to characterize fungal diversity on coral reefs more conserved markers such as the SSU or LSU rRNA might be recommendable. However, the overall high degree of detected fungal novelty warrants a need for fungal isolation efforts from corals to fill the gaps in the sequence databases. Finally, the fungal-specific primers from Nikolcheva & Bärlocher (2004) used in this study failed to prevent amplification of non-fungal sequences, a problem also encountered by Amend, Barshis & Oliver (2012) on the LSU marker.

Inferences from network analysis

The network of the combined Symbiodinium, bacterial, and fOTU tables resolved predominantly co-occurrence patterns involving bacteria. However, two co-occurrences were observed between bacteria and fungi. Co-occurrence of Symbiodinium A3 and an unclassified type A likely indicates either intragenomic variability within the A3 type or niche-overlap between closely related types. The network contained one sub-community, or module, consisting of six bacterial groups and connected by three negative correlations with a group of Planctomycetes of the class Phycisphaerae. The central node, having three positive correlations and one negative included bOTUs classified in the SILVA database to the unrecognized phylum PAUC34f. This enigmatic group of bacteria has not yet been cultured and sequences in the SILVA database are solely from metagenomic studies (Quast et al., 2013).

A noteworthy member of this node is the Chloroflexi SAR202 clade, as this includes the third most abundant bOTU. The node classified as Subgroup 11 of the Acidobacteria was connected to the PAUC34f node. Acidobacteria are specialized aerobic oligotrophs (Kielak et al., 2016) obtaining energy from denitrification and known to engage in syntrophic associations with different proteobacteria (Spring et al., 2000; Meisinger et al., 2007). Proteobacteria in the module were most similar to the genus Nitrosococcus and the family Nitrospinaceae, both of which are groups of nitrifying bacteria (Kowalchuk & Stephen, 2001; Lücker & Daims, 2014). Although little is known about the majority of organisms from this module, its members have distinctive metabolisms and it seems unlikely that the co-occurrence patterns are solely a result of niche-overlap.

Conclusion

In summary, the investigation of inter-domain microbial communities of S. siderea colonies from 17 ± 1 to 27 ± 1 m depth habitats at Conch reef revealed an exceptional diversity of Symbiodinium types, including types from seven of the nine currently recognized “clades” (Pochon, Putnam & Gates, 2014). While colonies at 27 m were dominated by an unclassified clade C type, colonies at 17 m were predominantly colonized by a mixture of clade B types. In contrast, the bacterial and fungal community compositions were not structurally different between the depth habitats, however, species indicator analysis revealed the presence of depth habitat-specific taxa. Further, we constructed a network to identify correlations within the microbiome, which reflected co-occurrence is most prevalent between bacterial groups, including a variety of taxa. Overall, our data show that the Symbiodinium community associated with S. siderea can be shifted drastically between different depth habitats within the same reef, which may indicate that alteration in this community is an important mechanism to physiologically adapt to local conditions. This work further presents evidence of a diverse fungal community associated with S. siderea, containing a high degree of novelty. It also emphasizes the need for future study on the diversity and functional roles of fungi within the coral holobiont and on the reefs in general.

Supplemental Information

Supplemental Information 1 Supplemental Figure 1.

Maximum likelihood phylogeny with the partial 16S rRNA gene showing the phylogenetic relation between bOTU1, the top BLAST hits and all SAR116 bacteria in the SILVA reference alignment, with the exception of KJ589655, HQ673528 and KF271096 which clustered outside of the SAR116 clade. SAR116 bacterial sequences acquired from sponges or corals are indicated in bold and with stars. All other SAR116 sequences are isolated from seawater. The optimal model was selected based on the AIC criterion and found to be GTR+G+I. The maximum likelihood analysis was conducted with RAxML, using a 1000 bootstrap iterations and contained 1,344 nucleotide positions (449 for bOTU1) with Rhizobium leguminosarum (ATCC 14480) as outgroup.

Click here for additional data file.

Supplemental Information 2 Supplemental Figure 2.

Maximum likelihood phylogeny with the partial 16S rRNA gene showing the phylogenetic relation between bOTU3 and clonal sequences from corals, Candidatus Lariskella, Candidatus Midichloria and other Rickettsiales genera. The optimal model (GTR+I+G) was selected based on the AIC criterion. The maximum likelihood analysis was conducted with RAxML, using a 1,000 bootstrap iterations and contained 1,176 nucleotide positions (452 for bOTU1) with Rhizobium leguminosarum (ATCC 14480) as outgroup.

Click here for additional data file.

Supplemental Information 3 MOTHUR Pipeline Analysis.

Click here for additional data file.

Supplemental Information 4 CP23S Reference Alignment.

Click here for additional data file.

Supplemental Information 5 cp23S Reference Classification FASTA file.

Click here for additional data file.

Supplemental Information 6 cp23S Reference Classification Taxa.

Click here for additional data file.

We are grateful to the Astronauts Luca Parmitano (ESA; European Space Agency), Serena Auñón (NASA), Norishige Kanai (JAXA; Japanese Aerospace Exploration Agency), and NASA Aquanaut David Coan for conducting the sample collections used in this study during the NASA Extreme Environment Mission Operations (NEEMO 20). We would also like to thank the Mission Directors Marc Reagan and Barbara Janoiko; and the entire NEEMO 20 team for their support and assistance during the coordination and planning of the marine science activities during the mission. We further would like to thank the Operations Director, Roger Garcia and the entire crew of the FIU Aquarius Reef Base for supporting and facilitating the diving and boating operations. We also express our thanks to Tanya Brown, Ellen Dow and Cindy Lewis from the IMaGeS Lab at FIU for assistance during the fieldwork. This is contribution #21 from the Marine Education and Research Center in the Institute for Water and Environment at Florida International University.

Additional Information and Declarations

Competing Interests

Author Contributions

Field Study Permissions

DNA Deposition

Data Availability

Mauricio Rodriguez-Lanetty is an Academic Editor for PeerJ. Trevor Graff and William Todd are employees of NASA Johnson Space Center.

Guido Bonthond conceived and designed the experiments, performed the experiments, analyzed the data, contributed reagents/materials/analysis tools, wrote the paper, prepared figures and/or tables, reviewed drafts of the paper.

Daniel G. Merselis conceived and designed the experiments, performed the experiments, analyzed the data, contributed reagents/materials/analysis tools, wrote the paper, prepared figures and/or tables, reviewed drafts of the paper.

Katherine E. Dougan conceived and designed the experiments, performed the experiments, analyzed the data, contributed reagents/materials/analysis tools, wrote the paper, prepared figures and/or tables, reviewed drafts of the paper.

Trevor Graff performed the experiments, contributed reagents/materials/analysis tools, reviewed drafts of the paper.

William Todd performed the experiments, contributed reagents/materials/analysis tools, reviewed drafts of the paper.

James W. Fourqurean contributed reagents/materials/analysis tools, reviewed drafts of the paper.

Mauricio Rodriguez-Lanetty conceived and designed the experiments, performed the experiments, contributed reagents/materials/analysis tools, reviewed drafts of the paper.

The following information was supplied relating to field study approvals (i.e., approving body and any reference numbers):

The National Oceanic and Atmospheric Administration, Office of National Marine Sanctuaries Program (ONMS): FKNMS-2015-076-A2.

The following information was supplied regarding the deposition of DNA sequences:

The sequence reads generated in this study have been deposited in the SRA database under the accession number PRJNA356144.

The following information was supplied regarding data availability:

The DNA data was deposited in NCBI database: https://www.ncbi.nlm.nih.gov/bioproject/PRJNA356144/

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
