# Peer review of "Inter-domain microbial diversity within the coral holobiont Siderastrea siderea from two depth habitats"

_PeerJ, doi:10.7717/peerj.4323_

## Round 0.1 · original submission · Major Revisions

Both reviewers agreed that major revisions are needed to reconsider this manuscript. In particular, please pay attention to the comments about not over-interpreting the results and making the findings more clear and concise. Please acknowledge limitations of this study and minimize speculation.

Reviewer 1 ·

Basic reporting

no comment

Experimental design

see detailed comments

Validity of the findings

see detailed comments

Additional comments

This study investigates the Symbiodinium, bacterial and fungal diversity of 14 Siderastrea sidera coral colonies sampled from one reef across two depths using amplicon sequencing approaches. The study is interesting in many aspects; though it also needs extensive revision in many areas. There are a number of points listed below that should be addressed in a revised manuscript, mainly focused on streamlining and shortening which will help to remove much of the speculation. It is recommended to focus on the novel aspects of the study, namely the combined analysis across the multiple components of the coral holobiont. However care should be taken to not over-interpret the results and where there are limitations these should be highlighted. The comments below are hoped to help in a revised manuscript.

Major Points:
• The manuscripts is very long for what is actually presented; i.e. amplicon data across 14 sampled colonies. While the combination of Symbiodinium, bacterial and fungal diversity metrics is a nice approach, and combined with network analysis brings in aspects of co-occurrence patterns, caution should be taken in interpretation of the patterns. There is a lot of speculation on adaption and functional roles of the associated microbial community without any actual functional evidence. This speculation contributes somewhat to the length of the paper and could be reduced to help streamline the main messages.
• It would be good to provide more details about the local conditions and habitat from which the corals were sampled from. It is stated that the samples are derived from 1 reef and 2 depths @ 17m and 27 m; however I find it difficult to see how just a 10 m depth difference can be driving many of the patterns and shifts in the microbes and how they are contributing to adaption of coral colonies as is stated in the manuscript. More information regarding the individual location of colonies on the reef would be advantageous and also importantly if possible any local habitat conditions and microenvironment water quality parameters that maybe influencing the 17 vs. 27 m coral colonies;
• The groupings postulated for the two depths in the Symbiodinium community as highlighted in the MDS plots (figure 2) seems not to be supported – as the Symbiodinium community for colony 2 is closer to that of Colony 12 than the other corals in this deeper cluster. Hence while there may be weak statistical support; based on the level of replication and inherent biases of amplicon approaches it is questioned how robust these conclusions are. One aspect not considered that may have an even greater influence on all microbial partners including the Symbiodinium populations is the coral host genotype. This should at least be discussed if no data is obtained for these genotypic differences.
• Inherent biases associated with the technology exist; i.e. sample to sample variation is in part a result of library prep and amplification efficiency. Duplicate or replicate samples from each individual colony would help identify the bias associated with these approaches. While this may be a hindsight detail – it is kind of important to provide greater rigor around if there is a real difference (for Symbiodinium) or lack thereof (for bacteria, fungi etc.) across the individual colonies at the two sampling depths.

Other points:
• Abstract – last sentence: How is local adaption assessed; There is no information in the manuscript around adaption or traits associated with adaption or functional information for the microbial associates. Hence unless some information is provided on different conditions the coral colonies experience at the two depths; inference to adaption should be removed.
• Line 51: This sentence starting “ Most well studies…..” is reads strange – perhaps rephrase
• Line 58 – provide a reference for “host flexibility”
• Sentence starting line 64 can be removed – adds little and “significant impact” is a strange way to refer to microbial associations with important functions to the coral holobiont.
• Line 77: Update other recent papers in this space; Agree there are not many but more are coming; i.e. Ainsworth eta l 2017;
• Line 81; This paragraph could fit better as the first paragraph of the introduction in some respects: It goes very broad again;
• Line 94: Sentence starting “ In addition….” Is repetitive – remove
• Line 126: remove “over the last two weeks”
• Line 128-132: Were coral colonies washed to remove loosely attached microbial associates and even microbes derived from the water column? What is the carry-over of contaminating microbes especially as corals were collected by what seems to be divers focused on other tasks as well? This is relevant to the dominant bacterial OTUs i.e. a SAR116 and Synechococcales affiliated OTUs. Both these OTUs also prevalent in water surrounding corals and more commonly documented from these water habitats. This needs to be discussed in the manuscript.
• Overall the discussion seems very long and could be cut back extensively to make the manuscript more focused.
• Line 316: The use of the term “subcomponents’ to describe the Symbiodinium, bacterial and fungal communities seems confusing to me; from my point of view this could also refer to different microhabitats of corals. Hence I would rephrase; i.e. the Different microbial populations
• Line 335: It could be questioned what is the relevance of depth to Symbiodinium diversity; in my mind such a narrow environmental difference may have limited influence; The other studies of Bongaerts et al for example are over much bigger depth gradients. Again the question of genotype seems more relevant here and needs discussion.
• Line 382: Very weak sentence here? Remove?
• Line 388 – remove remarkable? Avoid over descriptive language.
• Throughout the section discussing bacteria – i.e. lines 385 onwards; much of it is based on inferring potential functional roles of identified OTUs; This is very speculative and should be reduced where possible;
• Line 406 Sentence starting : The high abundance…” does not add much;
• Line 410: Certain OTUs can be signs for concern of the data; for example Ralstonia is a common contaminant in Bacterial amplicon sequencing studies. Were any negative controls included in the analysis; to provide a based line for contaminant sequences that are prevalent in all amplicon studies?
• Line 441: This sentence is not clear to me; what is it trying to say? Again much of the fungal discussion from line 421 can be reduced.
• Line 480-484: Is this paragraph providing anything? Remove?
• Line 526: remove – there is no data to support local adaption and so this conclusion cannot be supported.
• Much of the conclusion is repeating aspects of the paper rather than providing a summation of the major conclusions. For example lines 535-541 can be removed.

Reviewer 2 ·

Basic reporting

1. The manuscript is mostly clear, but there are a few occasions where meaning was obscured due to the lack of sentence structure. I have provided several examples of this and I recommend that the authors re-structure or break-up these sentences so that they are easier to read and interpret. See lines 157-158, 198-202, 236-238, 289-291, 339-343, 359-361, and 530-532. I would also recommend that the authors read over the manuscript closely- I noticed grammatical errors throughout the text.
2. The manuscript needs a conclusion paragraph within the discussion to tie everything together. Currently, the discussion ends abruptly with a paragraph concerning the abundant bOTUs.
3. The sentence “However, these hypotheses are poorly explored (lines 90-91)” needs to be revisited. This sentence suggests that very little to no research has been conducted concerning how shifts in the microbiome by depth may contribute to the ecological success of corals, but there are a number of articles that have attempted to address this. Some of these citations are already included in the paper.
4. The authors offer a Mothur script and other files within their supplementary materials (line 168, line 174), but I cannot see these files uploaded. Was this submitted with the manuscript?
5. When an object (i.e. reference database, software, resource) is introduced for the first time in the manuscript, the complete name of the object should be spelled out. There are several instances where the authors refer to the tool or resource with just the acronym (see lines 173, 184, and 186 for examples – this is not a comprehensive list and the authors should amend this if they revise the manuscript).
6. The abbreviation sOTU is not introduced with the term it is referring to (line 227) and this is confusing.
7. The use of the Basic Local Alignment Search Tool (BLAST) server should also be cited.
8. Adding more background about the physiology and life history (i.e. mode of reproduction, symbiont vertical transmission) of S. siderea would improve the manuscript (perhaps within the introduction?). Also, was there an ecological reason as to why S. siderea was selected for this study? If so, please provide this justification within the manuscript. Perhaps this information can also be used within the discussion to explain and interpret the results. Could this information also explain differences that are observed in Symbiodinium by depth?
9. Table 1: the table description should be written in complete sentences and references Deep (D), Middle (M), and Shallow (S) depths that are not within the table. The sentence “When types, fOTUS or bacterial families . . .” tells the reader that somehow information from the network analysis is included in the table, but this information is not in the table. The word “ Symbiodinium” is missing from this sentence as well.
10. The authors should be consistent with how they refer to the different microbial communities throughout the text. This is easier for the reader to interpret.
11. Figure 2: Please indicate how these polygons were drawn. Is this a statistical procedure?
12. Figure 3: Please add information about what the emboldened line represents and what the edges of the box represent (i.e. first or third quartiles).

Experimental design

1. The authors mention a number of times that differences in the environment may drive differences in the Symbiodinium community between depths, but no information about the environment is given at either depth, other than the location. To improve the strength of the study, the authors should provide this data (if they can) and its analysis and use this to develop clearer and more meaningful conclusions about their results and some of the influences that potentially drive differences in the Symbiodinium community.
2. Was the DMSO-EDTA solution used to collect the corals sterile (line 130)? How was it sterilized?
3. Was the seawater decanted from the initial collection tube immediately after ascent?
4. How were the coral pieces stored until they could be preserved?
5. I commend the authors for justifying their choice of reference database!

Validity of the findings

In reference to the indicator species analysis, the results stated in the text (lines 289-290) do not match the results in table 1; 3 bacterial OTUs were identified for either of the 2 depths. Reading from table 1, it appears that RSB-22 was only an indicator at 27m while the unclassified Alphaproteobacterium-III and Desulfovibrio were indicators at 17 m.

Additional comments

The manuscript “Interdomain microbial diversity within the coral holobiont Siderastrea siderea from two depth habitats” by Bonthond and colleagues describes the diversity and composition of the Symbiodinim, bacterial, and fungal communities associated with Siderastrea siderea corals over a narrow depth gradient (17 and 27 m). This study also introduces and provides new information to the field of coral microbiology in a number of ways. First of all, this manuscript provides information about the microbiome of S. siderea, which is an abundant Caribbean coral that has been relatively understudied with regard to its microbiome. Secondly, the manuscript characterizes the fungal community associated with these corals using amplicon sequencing and by doing so, provides methods that other researchers can use to target these communities as well as outlines limitations related to non-specific priming and database coverage that need to be addressed to advance research in this area. Thirdly, the manuscript attempts to look at co-occurrence patterns between Symbiodinium, bacteria, and fungi using network analysis to reveal potentially meaningful associations between microorganisms within S. siderea.
While the manuscript contains information, methods, and insight that will be useful to the field, the manuscript requires significant revisions. Overall, the manuscript lacks synthesis of the results within the context of what we already know about Symbiodinium and their importance within the coral host and this is most obvious within the discussion of the results. For example, the discussion never revisits the content stated within the conclusion sentence that is offered in the abstract “In terms of differences in microbial composition between the studied depth habitats, our data suggest that local adaptation of the S. siderea holobiont is achieved in parallel with a shift in the endosymbiotic Symbiodinium community while bacterial and fungal community compositions undergo little change.” After reading this discussion and seeing the results, I am not convinced that local adaptation (to what: depth? light availability? temperature?) is the cause for shifts in the Symbiodinium community. I have also provided several examples of this in the other comment sections.

---

## Round 0.2 · Minor Revisions

Thank you for resubmitting this email. Please address the comments by Reviewer 1, especially adding a section about the limitations of the study, shortening the manuscript (the reviewer suggested potential areas to reduce), and providing the missing script in the supplemental materials.

Reviewer 1 ·

Basic reporting

These aspects of the manuscript are OK (though scripts files missing in Supplementary Material)

Experimental design

Adequate with limitations highlighted below

Validity of the findings

Satisfactory though some qualifications are required

Additional comments

This revised manuscript details the Symbiodinium, bacterial and fungal diversity of 14 Siderastrea sidera coral colonies sampled from one reef across two depths using amplicon sequencing approaches. The revision has improved the manuscript and while most concerns have been addresses adequately, there are still some issues raised and some additional edits that would aid the manuscript when published. Overall I think the manuscript is acceptable for publication though it is limited in some aspects, though also it is acknowledged that most studies have some limitations.
The limitations include;

1. The limited number of samples in the study and the absences of within sample replicates to assess variation and biases that are inherent in the amplicon sequencing approaches.
2. The lack of any physio-chemical measurements of the environmental conditions (including light) but which are inferred to be structuring the Symbiodinium shifts.
3. The sampling methodology does raise issues for me with corals only being removed from the surrounding seawater and not actively washed to remove loosely attached microbes. Especially since these samples were collected by divers not specifically trained in sample collection and the samples spent up to 4 hours in the seawater before processing. All these factors can change the microbial community diversity patterns.
4. Finally I still think the manuscript is very long for a diversity analysis of 14 samples and suggest a few areas that can be further cut or reduced to focus the manuscript.
It would be advantageous if the Authors could provide some short qualification in a revised manuscript addressing the points 1 -3 above and also still address point 4 which does result in a lack of focus of the manuscript. These are only minor corrections that I believe would help the manuscript and provide a useful contribution to the coral microbiology field.

Other points:
• Lines 281- 303 is essentially discussion and not results. This section should be integrated into the discussion. It is still quite speculative assigning location to the coral tissues specifically based on retrieval of only sequences, especially considering the samples were not washed to remove loosely attached and mucus associated communities. It is also not clear why OTU5 sequences are specifically highlighted as presumably many of the recovered sequences are tissue associated, and or mucus, skeleton. This is an issue for many coral studies as the localized habitats of coral associated microbial communities are vague, hence based on these samples and only recovering sequences it is impossible to infer if they are surrounding, mucus, tissue, skeleton or gastric cavity associated microbes. I would recommend being very cautious in speculating on location without additional evidence.
• Lines 320-333 Can be removed as these two paragraphs are very short and essentially saying what is also discussed in the following paragraphs (i.e. line 334 onwards). This again contributes to the length and lack of focus in the manuscript.
• Lines 382-394; This section again is highly speculative based on recovery of a short region of the 16S rRNA gene sequence. Hence starting the bacterial discussion on such a speculative topic is problematic in my view. The discussion should initially focus on the results that can be supported – i.e. diversity patterns between sample depths and no significant changes (unlike the Symbiodinium). Then discuss some of the co-occurrence patterns and then bring in the speculation (in a very limited and focused way on this OTU1 combining the results (lines 281-303) with this paragraph. However keep it short and focused and do not go into too much unsubstantiated speculation.
• Note the Mothur script I can not see in the uploaded Supplementary files.

Reviewer 2 ·

Basic reporting

The manuscript is greatly improved in comparison to the first version of the manuscript.

Experimental design

I have no further comments on the experimental design.

Validity of the findings

I have no further comments on the validity of the findings.

Additional comments

There are a few grammatical issues in this manuscript that should be corrected in the copy editing process.

---

## Round 0.3 · accepted · Accept

Thank you for thoroughly addressing the reviewers' concerns.